# Field Investigation of Effect of Plants on Cracks of Compacted Clay Covers at a Contaminated Site

**DOI:** 10.3390/ijerph19127248

**Published:** 2022-06-13

**Authors:** Yu-Zhang Bi, Xian-Lei Fu, Shi-Ji Zhou, Jin Ni, Yan-Jun Du

**Affiliations:** 1Laboratory of Urban Underground Engineering & Environmental Safety, Institute of Geotechnical Engineering, Southeast University, Nanjing 210096, China; biyuzhang@seu.edu.cn (Y.-Z.B.); fuxianlei@seu.edu.cn (X.-L.F.); shijizhou@seu.edu.cn (S.-J.Z.); nijin@seu.edu.cn (J.N.); 2School of Civil and Environmental Engineering, Nanyang Technological University, Singapore 639798, Singapore

**Keywords:** contaminated site, compacted clay cover, field test, CCC crack, herbaceous plants

## Abstract

Compacted clay covers (CCCs) are effective in restricting the upward migration of volatile organic compound (VOC) and semi-volatile organic compound (SVOC) vapors released mainly from unsaturated contaminated soils and hence mitigate the risks to human health. Desiccation cracking of CCCs would result in numerous preferential channels. VOC or SVOC vapors can prefereially migrate through the cracks and emit into the atmosphere, exposing threats to human health and surrounding environmental acceptors. This study presented results of comprehensive field investigation of desiccation crack distribution in CCCs, where four herbaceous plants were covered at the industrial contaminated site in. The plants included *Trefoil*, *Bermuda grass*, *Conyza Canadensis*, and *Paspalum*, and the corresponding planting areas were labeled as S1, S2, S3, and S4, respectively. The quantity and geometry parameters of the cracks including crack width, depth, and length, were investigated. The results showed that the cracks of the CCCs were mainly distributed in the areas of S3 (*Conyza Canadensis*) and S4 *(Paspalum*), where more cracks were formed when the degree of compaction (DOC) of the CCCs was less than 87%. In addition, the results revealed that: (1) no cracks were found in the area S1 (*Trefoil*); (2) the quantity, average width, average depth, average length, and maximal length of the cracks in the investigated areas followed S4 (*Paspalum*) > S3 (*Conyza Canadensis*) > S2 (*Bermuda grass*); (3) the maximal crack length in the area S2 (*Bermuda grass*) was the shortest, which was approximately one-seventh and one-eighth of those in the areas S3 (*Conyza Canadensis*) and S4 (*Paspalum*), respectively; and (4) the maximal width and depth of the cracks followed S3 (*Conyza Canadensis*) > S4 (*Paspalum*) > S2 (*Bermuda grass*).

## 1. Introduction

Over the past few years, global climate change has led to frequent extreme arid climates, imposing water loss, and the shrinkage of clayey soils. As a result, the soil mechanical properties and hydraulic conductivity would change considerably [1,2,3]. Degradation of mechanical properties of clayey soils may induce engineering disasters, e.g., landslides and dam breaks [4,5,6,7,8,9]. Compacted clay covers (CCCs) are extensively used as engineered barriers at industrial contaminated sites to control vapor intrusion, i.e., upward migration of VOC and SVOC vapors across the contaminated soils in vadose zones and emit to atmosphere environment [10,11]. Previous studies showed factors affecting migration of VOCs included advection, diffusion, sorption/desorption, and degradation processes [12,13]. Diffusion has been identified as one of the dominant processes for vapor-phase VOC/SVOC transport in soil covers [14,15].

Studies on soil cracking focused on the cracking formation mechanism. Morris et al. [1,16] proposed relationships between the elastic modulus, Poisson ratio of soils, and the degree of soil cracking. Weinberger [17] investigated the complete evolution process of water losing and cracking of soil. Tang et al. [18] investigated the effects of clay content, dry-wet cycle, and fiber content on soil cracking. They proposed a method to characterize crack development. The shrinkage of clayey soils resulted from the evolution of suction during the soil water loss process. Soil physicists initially developed the general concept of suction in the early 1900s. The total suction consists of two components, namely, the matric and osmotic components [19]. Tension stress is exerted inside the soil if the shrinkage deformation is restricted, thus cracks is induced when the tension stress exceeds the tensile strength of the soil [20,21]. Oleszczuk et al. [22] verified that the overburden stress had close relationships between the clayey soil shrinkage geometry factor and moisture content, yielding lower values of desiccation crack volume than that without overburden stress. Fundamentally, the formation of desiccation cracks can be explained from micro and macro perspectives: (1) macroscopically, variations in the effective stress and volume contraction, and (2) microscopically, the interparticle skeletal and capillary forces and particle displacements. Clayey soils are particularly susceptible to volume change and desiccation cracks due to their high suction potential [23,24,25].

Studies on the effects of plants on soil cracking were initially started in agronomy [26]. Sharma and Verma [27] indicated that cracks would be easily induced on the surface of clayey soil at row-planted crop sites. Johnston and Hill [26] suggested that when a plant root was growing, the cracks in soils could be generated at positions with the following characteristics: (1) positions with minimal resistance to plant root induced deformation in soil, and (2) a relatively high-water content under the root extension path. Subsequently, Johnson [28] found that the extent of cracking was elevated considerably with an increase in plant row spacing. Fox [29] indicated that roots pushed the soil during plant growth, exerting external stresses on the soil, and therefore caused soil cracks. Previous studies have shown that plant roots and water migration in the soil significantly affect the soil cracks, and row spacing and plant root length were key factors [27,30,31,32,33]. Researches suggested that transpiration of plants could cause an uneven distribution of suction in soils and then yield soil cracks [34,35,36]. Plant roots may penetrate into the CCCs of municipal solid landfills, resulting in cracks and consequent rainfall percolation preferential channels in CCCs [37,38].

The aforementioned studies showed plants had promoted cracking of clayey soils. On the other hand, plants could enhance the anti-erosion capability of soils, mitigating hydraulic erosion in soils [39]. Herbaceous plants are more effectively and widely applied in geotechnical engineering than woody plants [40,41,42,43]. This is because herbaceous plants are easier to survive in heavily compacted soils. However, whether herbaceous plants can restrain cracking in CCC remains controversial. Some researchers found that the root growth of herbaceous plants could restrain the self-healing property of cracks. The transpiration of herbaceous plants could promote uneven distribution of suction stress in clayey soils. Thus, the presence of herbaceous plants may result in more cracks in soils [35,44]. In contrast, other scholars hold a different view where the roots of the herbaceous plants in soils could act as fiber reinforcement and consequently restrain soil cracking [45].

This study presents a field investigation of cracking in the CCC. The CCC was used diffusion barrier against upward migration of VOC and SVOC vapors at a industrial contaminated site located in Southeastern China. A thick CCC was constructed on the top of the VOC and SVOC contaminated soils, and different plant species were grown on the CCC. When exposed to the relatively high seasonal atmospheric temperature, the water in the CCC was gradually lost, and cracks occurred. This study (see graphical abstract) was conducted to explore the factors affecting crack generation. Firstly, the crack degree with different plant species was investigated in field studies. Secondly, laboratory tests were conducted to verify the relationship between the DOC and the crack degree. Thirdly, effects of plant species and DOC of CCCs on cracking were discussed. 

## 2. Field Tests

### 2.1. Vegetation Coverage

The distribution of herbaceous plants at the site is illustrated in Figure 1. Land A was primarily covered with *Trefoil*. There was a rectangular area with 240 m^2^ covered with small patches of *Conyza Canadensis*. In the west of Land A, there were four patches of *Paspalum* with an area of nearly 1060 m^2^. Land B was covered with three different plants including *Trefoil*, *Bermuda grass*, and *Conyza Canadensis*. To the west of Land B was a rectangular area covered with *Conyza Canadensis*. The center of Land B was an irregular boot-shaped area covered with *Trefoil*. To the east of Land B was a ladder-shaped area covered with *Bermuda grass*. Land C is almost overwhelmingly covered with *Bermuda grass*. The central oval patch (Land A) is covered with *Trefoil* with an area of approximately 140 m^2^. The CCC was covered by four types of plants including *Trefoil*, *Bermuda grass*, *Conyza Canadensis*, and *Paspalum.* These four corresponding areas were labeled as S1, S2, S3, and S4, respectively (see Figure 1).

### 2.2. Site Description

The test site was located in the southeast of China. The total area was nearly 26.2 hectares. The contaminated sites, namely Land A, Land B, and Land C, are illustrated in Figure 1. The size of Land A was approximately 18.7 hectares. The target contaminants in the soil of Land A included carbon tetrachloride, total petroleum hydrocarbon (TPH), chlorobenzene, dichlorobenzene, methylbenzene, and chloroform, among which the concentration of carbon tetrachloride was the highest when compared to other contaminants (see the Appendix A). Land B had an area of about 4.6 hectares. The target contaminants in the soil of Land B included chlorobenzene, dichlorobenzene, dichlorobenzene, and chloroform, among which the concentration of chlorobenzene was the highest compared to the other contaminants. The area of Land C was approximately 3.2 hectares. The target contaminants in the soil of Land C included chlorobenzene and TPH, among which the concentration of TPH was the highest.

Figure 2 shows the geological profile and soil properties at the contaminated site. It is seen that the ground soil strata could be divided into five layers: miscellaneous fill, clay, sandy silt, silty sand, and silty clay. The sandy content increased with the depth of the soil strata, whereas the clay content decreased. The natural water content of the compacted clay layer was 20–25%, which was lower than its liquid limit (38–41%) and higher than its plastic limit (14–17%). The organic content in layer (2) and layer (3) was 8%, twice as much as that of the other layers. The target contaminants in the unsaturated zone with different depths are shown in Appendix A. The primary contaminants in this site were Dichlorobenzene, TPH, chlorotoluene, chlorobenzene, and naphthylamine. 

There were three phases, namely, non-aqueous phase liquid (NAPL) phase, liquid phase, and gaseous phase of the NAPL contaminants, e.g., chlorobenzene, dichlorobenzene, 4-chlorotoluene, and chloroform in the unsaturated contaminated soils at the site. These contaminant vapors would migrate upward across the CCC due to advection and diffusion, whereas diffusion played a dominant role in the migration process [10,14,15]. The advection-diffusion flux of VOC or SVOC vapors would be greater when the cracks occurred. This is because the cracks had provided preferential channels for the upward vapor migration [46], since vapor or gas permeability and diffusion coefficient of the cracks were higher than those of the soil matrix of intact CCCs.

### 2.3. Properties of CCC

#### 2.3.1. Basic Properties

Practically planting soil with thicknesses of approximately 20 cm should be covered upon the CCCs. However, herbaceous plants were directly planted on the CCCs without covering the planting soil layer in this study. The reasons for this situation are as follows: (1) very limited time for the implement of this project, and (2) the upward migration of NAPL vapors had been effectively controlled with construction of CCCs, as it was observed in the field that unpleasant smell initially released from VOC and SVOC vapors in the atmosphere was not detected. Sampling of the in situ CCCs was based on ASTM D3441 [47], *Geotechnical engineering investigation handbook* [48] and some other references [49,50]. Sampling spacing was 50–70 m in the seriously contaminated area of Land A (located in the northwest and north area). The sampling spacing of 100–120 m was distributed in other regions. Figure 1 shows that the in-situ soils from 40 sampling points were finally obtained (see Figure 2) for the basic parameters’ tests. Sampling depth was one-half of the CCC thickness. The GXY-1 engineering driller was adopted for rotary drilling sampling. The methods of the drilling operation were employed as per the *Geotechnical engineering investigation handbook* [48], *Standard practice for classification of soils for engineering purposes* [51], and *Geotechnical design*, *Part 2*: *Ground investigation and testing* [52].

The CCC samples were sealed and packed after sampling. Firstly, the samples were enclosed in polyethylene bags. Secondly, the samples were put into iron boxes filled with buffer materials to prevent mechanical disturbance during transportation. Finally, all of the samples were transported to the laboratory for further studies. Table 1 shows the property parameters and test methods. According to ASTM D2487 [51], the soil of the CCCs was classified as CL.

#### 2.3.2. Degree of Compaction of CCCs under Different Vegetation Coverage

Measurement of degree of compaction (DOC)of CCCs under different vegetation coverage was conducted based on the cutting ring method. According to the Test Methods of Soils for Highway Engineering [57], the cutting ring method was used to evaluate the DOC of the fine soil that did not contain gravel. This method is widely adopted in laboratory and field tests due to its rapid process. 

Currently, there is no standard of classification of DOC of CCCs at contaminated sites. In this study, the Code for Construction and Quality Acceptance of Road Works in City and Town [58] was adopted to classify the DOC of the CCC at the test sites. The tested areas with a DOC ≥ 90% were classified as good compaction. With a DOC falling in the range of 87% to 90%, it was classified as interim compaction. The classification of DOC at the site is shown in Figure 3. It is seen that poor compaction was distributed in the areas of S3 and S4. The cracking was more severe in the area with lower DOC, whereas cracking was slight or non-existent in the area exhibiting higher DOC. 

### 2.4. Test Methods

#### 2.4.1. Degree of Vegetation Coverage

In the present study, the UAV (unmanned aerial vehicle) census method and quadrat method were adopted to investigate the vegetation coverage and crack distribution. This method could effectively enhance the efficiency of information collection in projects such as measuring and surveying the infrastructures and topography [59,60]. The UAV system is feasible to identify the vegetation and generate a detailed map of the vegetation assemblages at the species level [61]. The quadrat is a random sampling plot adopted to investigate the number of plant communities in agriculture and forestry research realms [62]. It was used in this study to ascertain the cracking of the CCCs at the site. First, the PHANTOM 4 PRO UAV was used to screen the vegetation coverage and obtain remote sensing images. Subsequently, the normalized difference vegetation index [NDVI)] data for the contaminated site were extracted from remote sensing images [63]. The fractional vegetation coverage (VFC) in the investigation area was then calculated by the method proposed by Matsushita et al. [64]:(1)$$\mathrm{VFC}=\frac{(\mathrm{NDVI}-{\mathrm{NDVI}}_{\mathrm{soil}})}{{(\mathrm{NDVI}}_{\mathrm{veg}}-{\mathrm{NDVI}}_{\mathrm{soil}})}$$
where ${\mathrm{NDVI}}_{\mathrm{soil}}$ denotes the NDVI value of bare soil or vegetation-free area; ${\mathrm{NDVI}}_{\mathrm{veg}}$ refers to the NDVI value of plants at the investigated site. The NDVI is calculated as per the following equation [65]:(2)NDVI=(NIR−RB)/(NIR+RB)
where NIR is the near-infrared band value and RB is the red band value.

Using the multiband sensors of UAV, the values of near-infrared band (NIR) and red band (RB) at the site were measured. The ranges of the NIR and RB values were 0.75–0.90 μ and 0.63–0.69 μ, respectively. The range in the NDVI value was [−1, 1]. The negative value revealed that the pixels of the remote sensing images in this area were clouds, water, or snow. When the NDVI is zero, the remote sensing images represent rock or bare soil. When the NDVI is positive, the remote sensing images represent plants. The NDVI value increases with increasing coverage degree of plants. 

Based on the classification of forestry coverage degrees [66], the VFC of 0–10%, 10–30%, 30–45%, 45–60%, and over 60% were classified as bare land, low coverage, medium-low coverage, medium coverage, and high coverage, respectively. As per Equation (1), the VFCs in S1 and S2 reached 92% and 86%, respectively, while only 44% and 31% in S3 and S4, respectively. Thus, the site was classified as a high VFC area (S1 and S2) and low VFC area (S3 and S4). 

#### 2.4.2. Crack Parameters of CCCs

The quadrat investigation method extensively used in the fields of forestry and agronomy [67] was applied in this study. Cracks were investigated using the quadrat method for areas with complete vegetation coverage (S1 and S2 in Figure 4). The quadrat area was 5 m × 5 m. The investigation steps in bare areas and low vegetation coverage areas (S3 and S4) were as follows: the target regions were first selected and survey points (see Figure 4) were taken randomly for the investigation with each point had the size of 5 m × 5 m (see Figure 4).

The distribution of cracks at each point was counted in Figure 4, and the parameters of the cracks including quantity, width, height, and length were recorded. The crack parameters were measured as follows: (1) the dividers were used to measure the length and width of cracks, and (2) the steps for measuring the depth included following steps: first, the No. 18 thin iron wire with a diameter of 1.2 mm and length of 35 cm with marked scale was chosen; subsequently, the thin iron wire was vertically inserted into the crack, and the length exposed to the air were measured and read. The crack depth, H, was obtained by minus the exposed length from the total length. The H was measured three times, and the average value was reported. 

## 3. Test Result Analyses

The distribution of cracks in the CCC is illustrated in Figure 5a–d. S1 exhibited the highest VFC and no cracks. S2 ranked second and showed a slight crack on the surface of the CCC. The cracks at S3 and S4 were more noticeable than those at S2. Subsequently, the quantity, length, width, and depth of the cracks in the areas were counted.

Figure 5 shows the crack investigation areas which are divided into Area A and Area B according to the value of the CCC compaction. The distribution of the DOC of CCCs is presented in Figure 3. Three points were taken for the crack investigation in each area with each point had a size of 5 m × 5 m. 

### 3.1. Effect of Plant Distribution on the Crack Parameters

Figure 6a presents the quantity of cracks in S2, S3, and S4. The quantity of cracks was the highest in S4 compared to the other areas. The quantity of cracks in area S4 was nearly two and eight times that in S3 and S2, respectively. It is seen from Figure 6a that the plant species significantly affected the quantity of the cracks. This is because the VFC in S4 was lower than that in S3, indicating that a wider area in the CCC in S4 was exposed to the atmosphere. In addition, a greater soil evaporation, i.e., faster water loss, was yielded in areas S3 and S4, because VFCs in S3 and S4 were lower as compared to S1 and S2. Therefore, cracks would more easily occur in S3 and S4. 

#### 3.1.1. Crack Length

Figure 6b shows the maximum crack length (MCL). The MCL values in S3 and S4 were 7.62 and 6.75 m, respectively. However, the MCL in S2 was only 0.92 m, one-eighth of S4 and one-seventh of S3, respectively. 

Figure 6c shows the statistical results of average crack length (ACL), suggesting that the ACL values of S4-A, S4-B, S3-A, and S3-B were 2.15 m, 1.62 m, 1.81 m, and 1.18 m, respectively. In contrast, the ACL values of CCC in S2-A and S2-B were only 0.21 m and 0.12 m, respectively. The ACL values were approximately one-ninth of S3 and one-tenth of S4, respectively.

#### 3.1.2. Crack Depth and Width

Figure 6d shows the data of maximum crack depth (MCD) and maximum crack width (MCW). In general, the MCD and MCW in Area A were higher than those in Area B. The MCD in S3-A was 25.1 cm and that in S4-A reached 22.6 cm. However, the MCD in S2-A was only 4.9 cm and was approximately 1/5 of that in S3-A and S4-A. The MCW in S3-A and S4-A was 4.5 cm and 3.1 cm, respectively, whereas the MCW in S2-A was only 0.8 cm, about 1/6 of that in S3-A and 1/4 of that in S4-A. Moreover, the MCD and MCW in S3-A were the highest compared to the other cases, which is attributed to the more developed roots of *Conyza Canadensis*. The maximum root diameter of *Conyza Canadensis* on the site was 5 cm, which was five times that of *Paspalum*. The root system can squeeze the surrounding soil during the growth of plants, thereby intensifies soil cracking and generate more profound cracks.

Figure 6e presents the statistical results of the ACD and ACW. The ACD varied in the order of S4 > S3 > S2. The ACDs of S4-A, S4-B, S3-A, S3-B, S2-A, and S2-B were 13.1 cm and 9.1 cm, 12.5 cm, 8.2 cm, 2.5 cm, and 1.8 cm, respectively. Likewise, the ACW in S4 was slightly higher than that in S3, which was significantly higher than that in S2.

### 3.2. Effect of Plant Distribution on Cracking

For a given thickness of CCC, a large crack depth results in a long preferential pathway for the migration of VOC/SVOC vapors. Therefore, the crack depth was taken as the criterion for the cracking levels. As the MCD was about 24 cm, the crack depth was divided into three levels, (i.e., 0–8 cm, 8–16 cm, and 16–24 cm), representing slight cracks, moderate cracks, and severe cracks respectively.

Figure 7 shows the frequency distribution of the cracking. The investigation point was classified as the slight cracking area where 0–8 cm crack depth was most frequently occurred, moderate cracks area where 8–16 cm crack depth was most frequently occurred, and severe cracks area where 16–24 cm crack depth was most frequently occurred. On this basis, cracking levels were ascertained at different plant covering areas, and distribution of the cracks and plants of the CCC at the site was shown in Figure 8.

Figure 8 presents the relationships between the cracking levels of the CCC and plant distribution. The cracks were primarily distributed in the areas covered with *Conyza Canadensis* and *Paspalum*. The crack degree was severe in S3 of Land B. However, the crack degree was moderate in the S3 of Land A. In S4 of Land A, the cracking levels were moderate in the west area, but severe in the central area. With the DOC distribution pattern shown in Figure 3, the following points can be drawn: CCC underwent severe cracks when the DOC was about 75%, and it experienced moderate cracks when the DOC was above 87%. Moreover, there was no crack for the S1, and only slight cracks existed in the southeast side of Land C for the S2. 

## 4. Discussion

### 4.1. Influence of Plants on Cracks

Field investigation revealed that the CCCs covered with *Conyza Canadensis* (S3 area) and *Paspalum* (S4 area) was more suspectable to severe cracks when the DOC was poor. At the same time, the CCC tended to exhibit moderate cracks when the DOC was interim. There were slight cracks for the area covered with Bermuda grass (S2 area) when the DOC was poor. There was no cracking in the *Trefoil* (S1 area) covered area. The reasons for the observations are discussed below. 

The influence of suction

The growth of *Conyza Canadensis* and *Paspalum* (in S3 and S4 areas) complies with the rule of row planting [27], suggesting that S3 and S4 possessed a higher spacing between adjacent plants, lower VFC values, and a larger area of bare soils, i.e., un-plated soils. As a result, the bare soil would easily loose water when exposed to relatively high-temperature conditions, causing a more remarkable difference in the water content of the CCC along the horizontal direction between the bare CCC and the CCC planted with *Conyza Canadensis* and *Paspalum*. As a result, the suction in the bared CCC is higher than that of the planted parts. In other words, the suction difference initiates along the horizontal direction of the CCC. As a result, cracks forms on the surface of the CCC in areas S3 and S4 [2,21]. 

2.The influence of root systems

Figure 9 shows the root morphology investigation of different plants. Figure 9a,b shows that the root of *Conyza Canadensis* and *Paspalum* is much more developed than that of *Bermuda grass* and *Trefoil* (Figure 9c,d). The root systems of *Conyza Canadensis* and *Paspalum* belong to the taproot system (Figure 9c,d), while the root systems of *Bermuda grass* and *Trefoil* belong to the fibrous root system. 

Figure 10 shows the schematics of the different root system in the CCCs. Figure 10a,b shows that the fibrous root system is distributed in the CCC with a high DOC, akin to adding fiber reinforcement in the soil. Figure 10c,d shows that the taproot system is distributed in CCC with a low DOC, which means that the CCC in this area can be divided into two sections: the reinforced section R1 and the un-reinforced section R2. The R1 section is difficult to crack because the root is reinforced, while the R2 section is prone to cracking. 

Previous studies have shown that plant roots may penetrate into the CCCs and generate cracks [37,38]. These kinds of roots are indeed suitable for the taproot system. This is because the taproot is much more robust than the lateral root and adventitious root (as seen in Figure 10e,f). Therefore, the diameters and length of the taproot of *Conyza Canadensis* and *Paspalum* were investigated. The survey area is shown in Figure 5. Figure 11 shows that the taproot sizes in the B area were larger than those in the A area. This can explain why more cracks were generated in B areas than in the A areas (see Figure 5). However, the taproot diameters of *Conyza Canadensis* were higher than *Paspalum*, while the cracks in the *Paspalum* area were more severe. This may be affected by some other factors such as DOC and water content of CCCs.

### 4.2. Influence of DOC on Crack

Previous studies by Wei et al. [68] showed that compacted expansion soil was closely related to soil cracking. Their results revealed that with the increase in the DOC of the soil, the degree of cracking decreased. Wei et al. [68] found the cracking usually occurred when the DOC was 75%, whereas the crack length of the soil with DOC of 75% was 3.5 times of soil with DOC of 85%. We conducted a series of tests to validate if DOC would considerably affect cracking of CCCs. The DOC values used to prepare the CCC samples were 75%, 87%, and 95%, based on the field investigation results. The CCC collected from the contaminated site was dried, mashed, and passed through a 2 mm sieve. According to the results in the above sections, the moisture content of the statically compacted CCC samples was controlled as 25.6%. The compacted CCC samples were immediately cured under standard conditions.

The test procedures are shown in Figure 12 as follows: (a) preparing the CCC samples using the static compaction method; (b) the CCC samples were heated at 40 °C for 24 h. Heating the CCC samples at 60 °C and 110 °C for 24 h were also conducted for comparison purposes; and (c) the image acquisition of samples was carried out by using an optical digital camera (Nikon D90) and lens (Nikon AF-S DX NIKKOR). Furthermore, to eliminate the influence of natural light, the image acquisition was carried out in a dark space and illuminated with an LED light; (d) the collected images were binarized and denoised using the CIAS software released from Tang’s research group at Nanjing University (www.climate-engeo.com, accessed on 8 March 2020).

To analyze the crack images, the following parameters were defined: (1) N is the number of crack nodes, and N_1_ is the number of cracks; (2) L is the sum of the lengths of cracks. The cracking of the CCC is shown in Table 2. When the DOC was 75%, the CCC cracked at 40 °C, while he CCCs with DOC degrees of 87% and 95% did not crack. When the temperature was 60 °C, more cracks occurred in the CCC with DOC of 75% compared to 40 °C. The number of nodes, crack numbers, and crack lengths were increased by 6 times, 6.5 times, and 1.95 times, respectively. Under the high temperature of 110 °C, the crack resistance of CCC with DOC of 95% was significantly superior to CCCs with DOC of 75%. The crack length in the CCC with DOC of 95% was reduced by 3.97 times that with DOC of 75%. CCC with DOC of 87% also showed excellent ability to inhibit cracking at 40 °C. Furthermore, a slight crack occurred at 60 °C, and the number of cracks was only 1/6 of the CCC with DOC of 75%.

## 5. Conclusions

This study investigated the desiccation cracks of the CCC at an industrial contaminated site in Southeastern China. The distributions of the cracks, plants, and degree of compaction of the CCC at the site were presented. The effects of the plants on cracks were discussed. Based on the results, the following conclusions can be drawn:(1)The cracking levels of the CCC were associated with the coverage of plants: the areas at the S3 (*Conyza Canadensis*) and S4 (*Paspalum*), the areas at S2 (*Bermuda grass*), and the area at S1 (*Trefoil*) had severe, slight, and almost non-exist cracking levels, respectively.(2)The number of CCC cracks in the areas at S4 was the largest, twice and eight times that of S3 and S2, respectively. The maximum crack length and average crack length of S2 were the minimum, which was approximately 0.92 m, and 0.21 m, respectively. Moreover, the maximum crack length of S2 was 1/7 that of S3 and 1/8 that of S4, respectively, and the average crack length of S2 was 1/9 of S3 and 1/10 of S4, respectively.(3)The maximum crack depth and maximum crack width of the areas at S2 were the minimum, 1/5 and 1/6 those of S3, and 1/5 and 1/6 those of S4. The average crack depth and average crack width at S2 were 1/5 and 1/6 those of S3, respectively, and 1/5 and 1/7 those of S4, respectively.(4)The cracking was closely related to the DOC of CCC: when the DOC was less than 87% at the S3 area, severe cracks occurred; when the DOC was greater than 87%, moderate cracks occurred. The areas at S4 also complied with the pattern.(5)The effect of plants on cracking was considerably exerted by the uneven distribution of the CCC surface suction caused by row planting. Furthermore, plants with fibrous root systems were beneficial for inhabiting CCC cracking.

## Figures and Tables

**Figure 1 ijerph-19-07248-f001:**
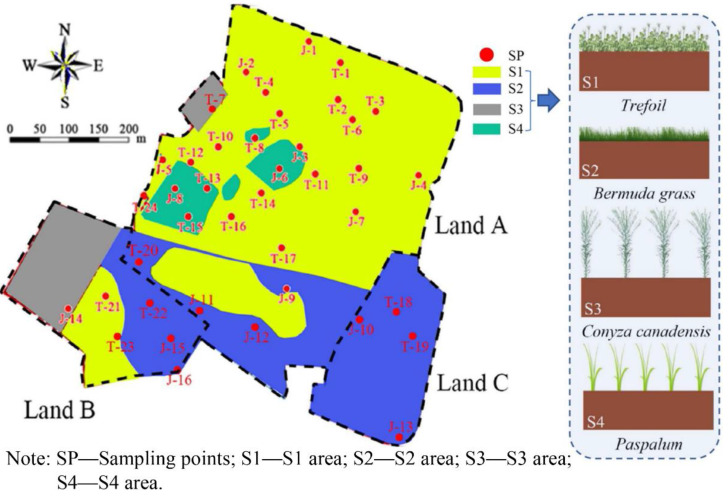
The soil sampling points at the test site and its vegetation coverage.

**Figure 2 ijerph-19-07248-f002:**
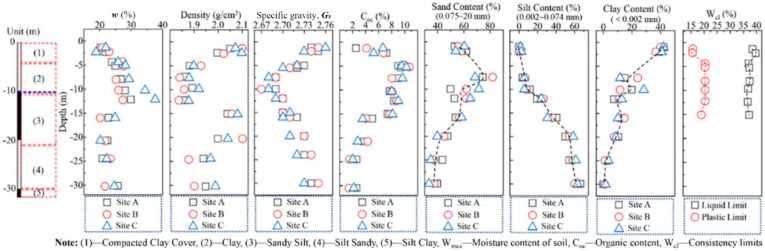
The geological profile and soil properties in the contaminated site.

**Figure 3 ijerph-19-07248-f003:**
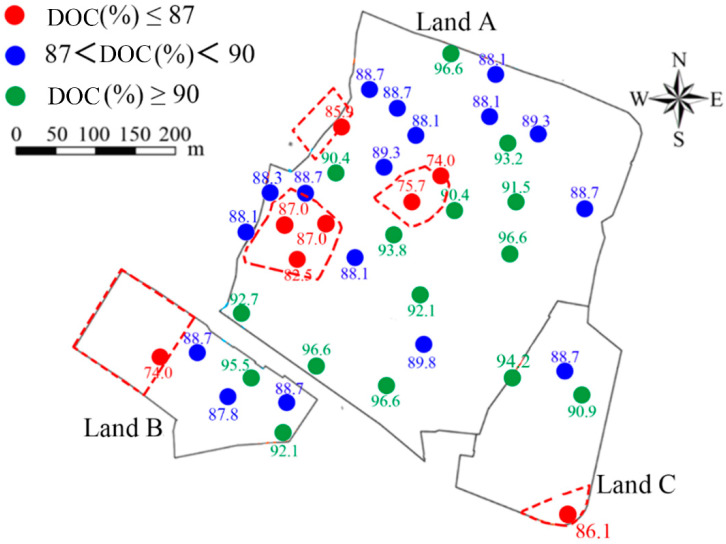
The degree of compaction (DOC) of CCCs at the testing site.

**Figure 4 ijerph-19-07248-f004:**
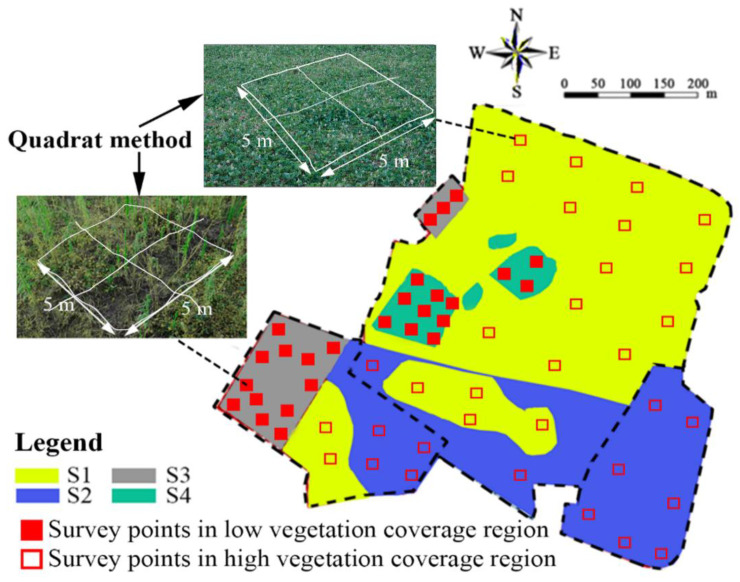
The selection of the investigation points for the CCC cracking.

**Figure 5 ijerph-19-07248-f005:**
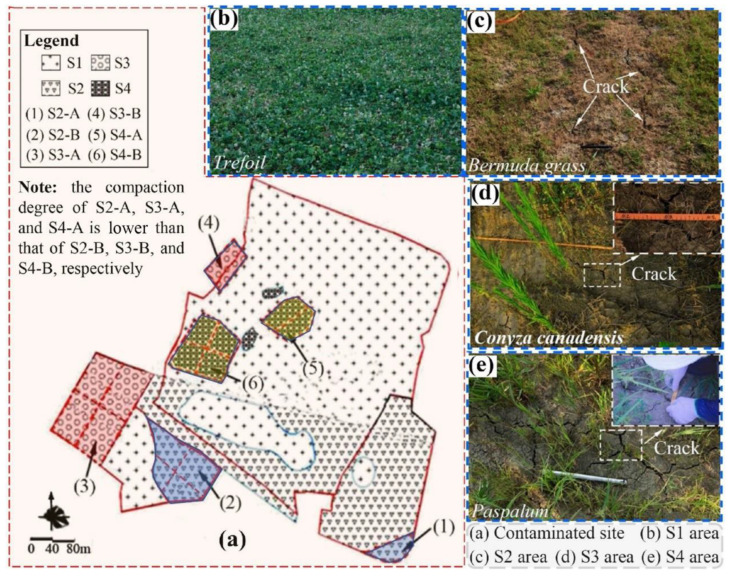
The schematic diagram of the cracking investigation under different vegetation coverage: (**a**) soil cracking investigation area was divided according to the degree of compaction of CCCs; (**b**) S1 (*Trefoil*) area; (**c**) S2 (*Bermuda*); (**d**) S3 (*Conyza Canadensis*); and (**e**) S4 (*Paspalum*).

**Figure 6 ijerph-19-07248-f006:**
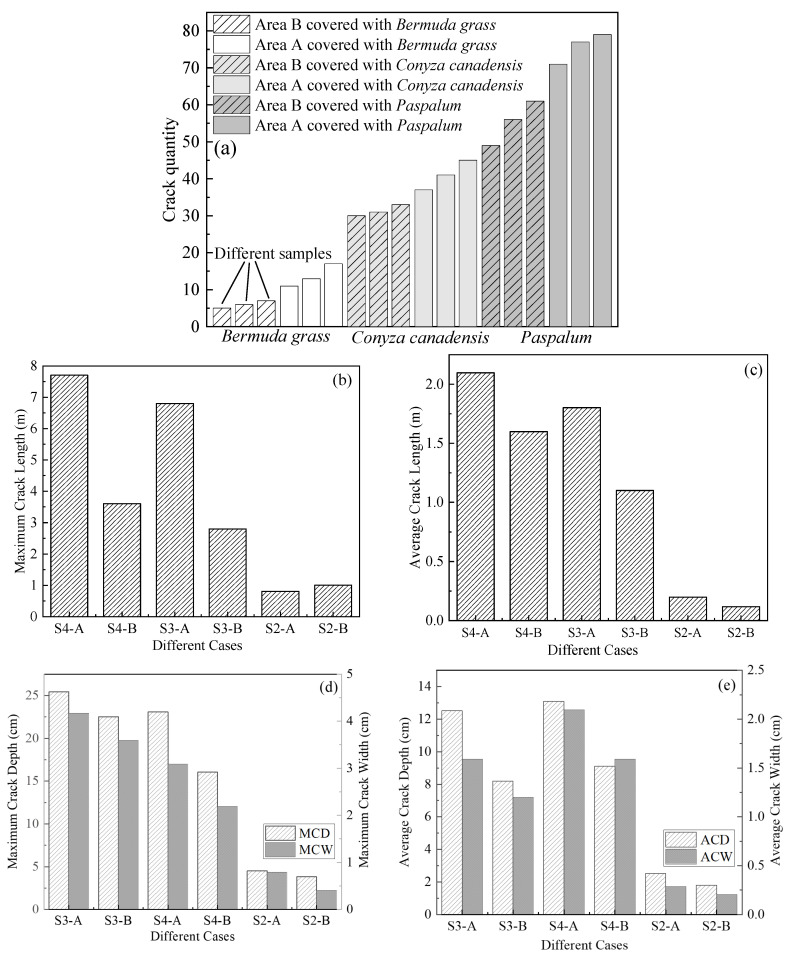
The results of the crack investigation: (**a**) the effects of the plant species on the quality of the CCC cracks; (**b**) the effects of the plant species on the maximum crack length of the CCC cracks; (**c**) the effects of the plant species on the average crack length of the CCC cracks; (**d**) the effects of the plant species on the maximum crack depth and maximum crack width of the CCC cracks; and (**e**) the effects of the plant species on the average crack depth and average crack width of the CCC cracks.

**Figure 7 ijerph-19-07248-f007:**
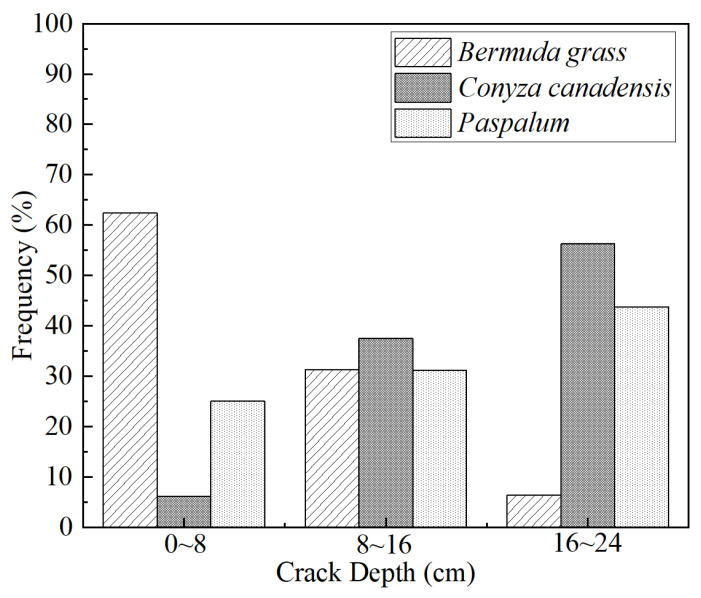
The frequency distribution of the CCC crack depths.

**Figure 8 ijerph-19-07248-f008:**
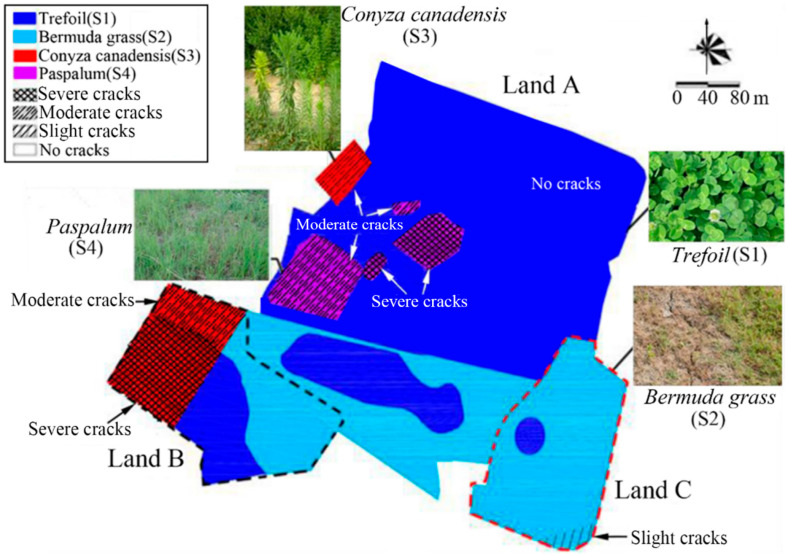
The schematic effects of plant distribution on crack distribution: the relationship between the crack degree and plant distribution.

**Figure 9 ijerph-19-07248-f009:**
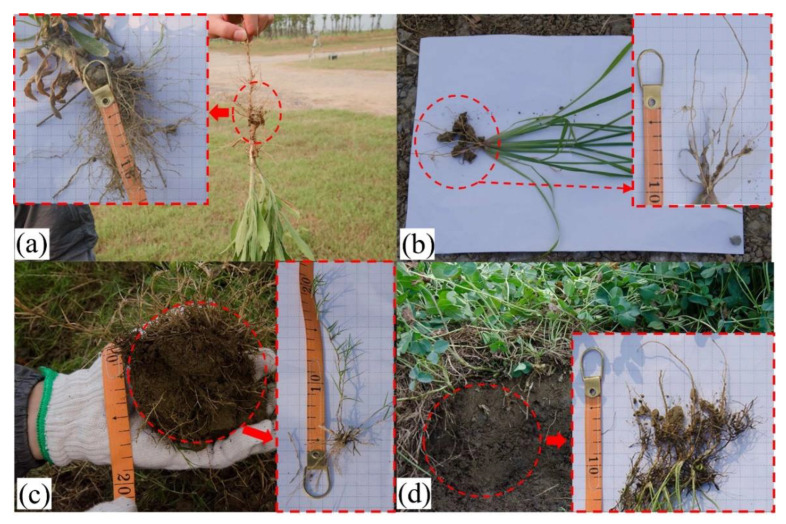
Photographs of the root systems in the field investigation: (**a**) *Conyza Canadensis*; (**b**) *Paspalum*; (**c**) *Bermuda grass*; and (**d**) *Trefoil*.

**Figure 10 ijerph-19-07248-f010:**
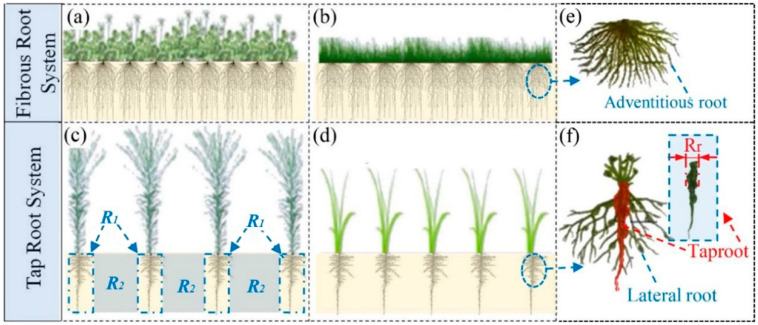
The schematics of the root systems of different plants: (**a**) *Trefoil*; (**b**) *Bermuda grass*; (**c**) *Conyza Canadensis*; (**d**) *Paspalum*; (**e**) schematic of fibrous root; and (**f**) schematic of taproot.

**Figure 11 ijerph-19-07248-f011:**
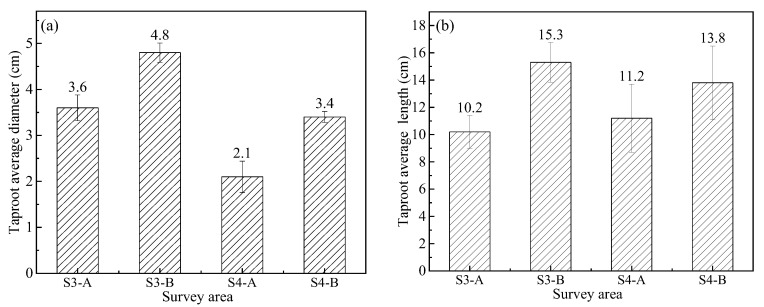
The taproot size survey in different areas in Figure 5: (**a**) average taproot diameter, and (**b**) average taproot length.

**Figure 12 ijerph-19-07248-f012:**
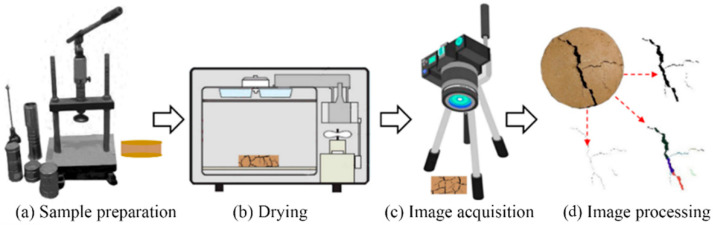
The schematic diagram of the test process.

**Table 1 ijerph-19-07248-t001:** The basic physical parameters of the CCC.

Property	Value	Geotechnical Test Methods
Moisture Content (%)	22.5	ASTM D2216 [53]
Specific Gravity, G_s_	2.72	ASTM D854 [54]
Liquid Limit, LL (%)	38.14	ASTM D4318 [55]
Plastic Limit, PL (%)	14.90	
Optimum Water Content (%)	25.6	ASTM D4253 [56]
Maximum Dry Density (g/cm^3^)	1.78	

Note: ASTM = American Society for Testing and Materials.

**Table 2 ijerph-19-07248-t002:** The results of the CCC crack measurement.

Temperature (°C)		40	60	110
N	75%	2	6	10
	87%	0	1	6
	95%	0	0	3
N_1_	75%	4	13	21
	87%	0	2	12
	95%	0	0	4
L (cm)	75%	1.1	2.2	6.6
	87%	0	0.9	3.1
	95%	0	0	1.7

## Data Availability

The data that support the findings of this study are available from the first author, Yu-Zhang Bi, upon reasonable request.

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
