# Peer review of "Field Investigation of Effect of Plants on Cracks of Compacted Clay Covers at a Contaminated Site"

_ijerph, 2022, doi:10.3390/ijerph19127248_

Round 1

Reviewer 1 Report

Review of the manuscript – ijerph-1731989

Field Investigation of Effect of Plants on Cracks of Compacted  Clay Covers at a Contaminated Site

The manuscript submitted for review presents the results of a study on the effects of concentration and species of selected plants on the distribution of cracks of compacted clay covers occurring in industrially polluted areas of China. The Authors determined the number and geometric parameters of the cover cracks through field studies. In addition to its scientific value, the work is also of utilitarian significance.  The Abstract, Introduction of the manuscript are well edited. The Authors have performed a comprehensive literature review of the research topic undertaken. They have demonstrated the validity of the research undertaken and have clearly stated its purpose. Conclusions relate to the purpose and topic of the research conducted. The Authors sufficiently described the investigated area, i.e. its geological profile, soil properties and vegetation cover. However, some need to be supplemented or corrected, namely:

  1. Please clearly state what is new in the conducted research, what is new?
  2. Serious doubts are raised by the lack of description of the morphology and degree of development of root systems (clumps) of plants, whose presence was taken into account in determining the size and geometry of cracks in the clay covers. This issue was omitted by the Authors, and yet the formulated aim of the work - page 3, lines 107-109 states: "The research steps .... were conducted to point out the effect of different plant species on crack generation". Different species, and therefore different plant morphology. This is important because already in the introduction to the manuscript, referring to relevant works of other authors and on page 10, lines 298 - 300, the Authors pointed out that the frequency of occurrence and geometry of these cracks may vary depending on the root system, and thus on the species of a given plant. Please complete this information.
  3. I think it is appropriate to include this issue in the discussion of the results.
  4. The first word of the Abstract should be corrected - Compacted Clay Covers.
  5. The caption of Figure 2 should be corrected - different typeface and Figure 10 - instead of "a" it should be "A".
  6. The title of the chapter "2 Effect of Plant Distribution on Crack Distribution" should be reworded.
  7. Comments on the References:
  8. the citation of other papers and standards should be corrected according to the guidelines of the journal
  9. the literature list lacks the item Johnson (1962), which the authors referred to on page 2, line 75
  10. also missing are GB 50021-2009; HJ 25.1-2014; JGJ 87-92; CJJ1-2008; JTG E40-200. The authors cited only ASTM standards.

Recommendations - major revisions

The manuscript still needs to be completed with the mentioned description of the plants, to determine the influence of their root system on the observed phenomena and to organize the cited literature.

Author Response

Response to Reviewer #1:

Above all, we would like to thank you for the specific and constructive comments on this manuscript. It has helped shape the work in all aspects and allowed us a useful examination of our work and the improvements that have been made.

Below are specific responses to the comments and criticisms.

Reviewer #1: Please clearly state what is new in the conducted research, what is new?

Response: A graphical abstract was drawn by authors to explain this question, furthermore, the relevant innovation points were listed in papers.

Reviewer #1: Serious doubts are raised by the lack of description of the morphology and degree of development of root systems (clumps) of plants, whose presence was taken into account in determining the size and geometry of cracks in the clay covers. This issue was omitted by the Authors, and yet the formulated aim of the work - page 3, lines 107-109 states: "The research steps .... were conducted to point out the effect of different plant species on crack generation". Different species, and therefore different plant morphology. This is important because already in the introduction to the manuscript, referring to relevant works of other authors and on page 10, lines 298 - 300, the Authors pointed out that the frequency of occurrence and geometry of these cracks may vary depending on the root system, and thus on the species of a given plant. Please complete this information. I think it is appropriate to include this issue in the discussion of the results.

Response: The discussion about root systems have been added in manuscript.

Reviewer #1: The first word of the Abstract should be corrected - Compacted Clay Covers.

Response: It has been modified in the manuscript.

Reviewer #1: The caption of Figure 2 should be corrected - different typeface and Figure 10 - instead of "a" it should be "A".

Response: It has been modified in the manuscript.

Reviewer #1: The title of the chapter "2 Effect of Plant Distribution on Crack Distribution" should be reworded.

Response: It has been modified in the manuscript as follows: Effect of plant distribution on cracking.

Reviewer #1: Comments on the References: (1) the citation of other papers and standards should be corrected according to the guidelines of the journal, (2) the literature list lacks the item Johnson (1962), which the authors referred to on page 2, line 75, (3) also missing are GB 50021-2009; HJ 25.1-2014; JGJ 87-92; CJJ1-2008; JTG E40-200. The authors cited only ASTM standards.

Response: The references have been modified in the manuscript.

Reviewer 2 Report

The manuscript entitled “Field Investigation of Effect of Plants on Cracks of Compacted 2 Clay Covers at A Contaminated Site” shows an interesting work, but the explanation of the data must be improved. The authors should explain the results and discussions more concisely. In addition, there are many typing and formatting errors that make it difficult to understand some sections.

When compared to the intro section, this latter is even too long and heavy to read, while the data are not clearly and properly reported. 
So, I would suggest reshaping the paper, making the intro easier to read, and discussing the data more thoroughly.

The experiments performed and the methodology seem to be appropriate for the conclusions, however, the discussion is more confused, maybe because of the poor English language. Therefore, I would like to review again the paper after the suggested reshaping. 

Author Response

The authors would like to thank the reviewers for your rigorous and methodical review. The comments have helped to improve our revised manuscript. To enhance the readability and impact of the paper, a professional English editor has been invited by authors to revise this manuscript. 

Round 2

Reviewer 1 Report

Dear Authors,

 All comments made in the review were taken into consideration and necessary corrections were made to the manuscript. Unfortunately, after the corrections were made, there were numerous errors in the numbering of bibliographic data in the References section that need to be corrected.

Recommendations - minor revisions

In content terms, the manuscript is no longer objectionable. Editorial corrections are necessary. In my opinion, after their improvement, the manuscript can be published.

Author Response

Thank you for your recognition of our works.

The reference part also have some mistakes. It has been revised by authors.

Reviewer 2 Report

The authors modified and implemented the manuscript as requested.

Author Response

(The authors gave the same response as above.)
